# New Insights into Cellular Functions of Nuclear Actin

**DOI:** 10.3390/biology10040304

**Published:** 2021-04-07

**Authors:** Malgorzata Kloc, Priyanka Chanana, Nicole Vaughn, Ahmed Uosef, Jacek Z. Kubiak, Rafik M. Ghobrial

**Affiliations:** 1The Houston Methodist Research Institute, Houston, TX 77030, USA; PChanana@houstonmethodist.org (P.C.); NVaughn@houstonmethodist.org (N.V.); Auosef@houstonmethodist.org (A.U.); RMGhobrial@houstonmethodist.org (R.M.G.); 2Department of Surgery, The Houston Methodist Hospital, Houston, TX 77030, USA; 3The Genetics Department, MD Anderson Cancer Center, The University of Texas, Houston, TX 77030, USA; 4Department of Regenerative Medicine and Cell Biology, Military Institute of Hygiene and Epidemiology (WIHE), 01-163 Warsaw, Poland; jacek.kubiak@univ-rennes1.fr; 5Dynamics and Mechanics of Epithelia Group, CNRS, Faculty of Medicine, Institute of Genetics and Development of Rennes, University of Rennes, UMR, 6290 Rennes, France

**Keywords:** nuclear actin, nuclear architecture, intranuclear rods, RhoA, F actin, G actin, chromatin remodeling

## Abstract

**Simple Summary:**

It is well known that actin forms a cytoplasmic network of microfilaments, the part of the cytoskeleton, in the cytoplasm of eukaryotic cells. The presence of nuclear actin was elusive for a very long time. Now, there is a very strong evidence that actin plays many important roles in the nucleus. Here, we discuss the recently discovered functions of the nuclear actin pool. Actin does not have nuclear localization signal (NLS), so its import to the nucleus is facilitated by the NLS-containing proteins. Nuclear actin plays a role in the maintenance of the nuclear structure and the nuclear envelope breakdown. It is also involved in chromatin remodeling, and chromatin and nucleosome movement necessary for DNA recombination, repair, and the initiation of transcription. It also binds RNA polymerases, promoting transcription. Because of the multifaceted role of nuclear actin, the future challenge will be to further define its functions in various cellular processes and diseases.

**Abstract:**

Actin is one of the most abundant proteins in eukaryotic cells. There are different pools of nuclear actin often undetectable by conventional staining and commercial antibodies used to identify cytoplasmic actin. With the development of more sophisticated imaging and analytical techniques, it became clear that nuclear actin plays a crucial role in shaping the chromatin, genomic, and epigenetic landscape, transcriptional regulation, and DNA repair. This multifaceted role of nuclear actin is not only important for the function of the individual cell but also for the establishment of cell fate, and tissue and organ differentiation during development. Moreover, the changes in the nuclear, chromatin, and genomic architecture are preamble to various diseases. Here, we discuss some of the newly described functions of nuclear actin.

## 1. Introduction

Nuclear actin was first described in bovine thymus cells [1,2], two different slime molds [3,4], Xenopus frog and newt Triturus oocytes [5,6,7], and liver cells [8]. However, these findings met with disbelieve and criticism [9], and after decades of skepticism and denial, finally around ten years ago, main-stream science not only accepted the existence of nuclear actin but appreciated its importance in the regulation of structure, function, homeostasis, and health of eukaryotic cells [10].

One of the reasons for the denial of nuclear actin existence was due to an inability to visualize nuclear actin using conventional phalloidin or antibody staining. Actin has three main polymerization states, monomeric (globular) G-actin, oligomeric, and polymeric (filamentous) F-actin. While the cytoplasmic actin is mainly monomeric or polymeric, nuclear actin is either monomeric, dimeric, oligomeric, or may adopt some other unusual conformations [11,12,13,14]. Nuclear actin oligomers, which cannot be stained by phalloidin, are often recognized by specific antibodies, such as the 2G2 antibody. This antibody recognizes the epitope composed of three nonsequential regions (aa131–139, aa155–169, and aa176–187) located in the vicinity of the nucleotide-binding region [15]. This indicates that nuclear actin adopts a specific conformation, absent in the cytoplasmic F-actin, which compacts these three separate regions into a single antibody-recognizable epitope [15,16]. Recent studies indicate that different antibodies recognize different pools of nuclear actin, with different conformation, and different locations within the nucleus, suggesting different functions [17]. Although our knowledge about the conformations adopted by nuclear actin is still very limited, there is a lot of information about the structural and functional roles of nuclear actin. We describe the role of intranuclear actin network and nuclear actin rods, and the role of nuclear actin in nuclear envelope breakdown, chromatin organization and remodeling, and transcription. For the description of the other roles of nuclear actin, such as apoptosis [18], viral infection [19], structure of nuclear membrane [20,21,22,23], we direct readers to the cited above papers.

## 2. Role of Intranuclear Actin Network

The presence of intranuclear actin monomers (unpolymerized G-actin) has been confirmed by many studies. The monomeric G actin constantly shuttles between the nucleus and cytoplasm. The Importin 9 protein imports G actin into the nucleus and the Exportin 6 protein mediates actin export to the cytoplasm [24,25]. These proteins function with the participation of the small GTPase Ran [25,26,27,28,29]. During the nuclear import, the GTP-bound Ran detaches cargo from the importin and returns importin to the cytoplasm. During the nuclear export, the nuclear GTP-bound Ran aggregates with exportin and cargo. This trimeric complex moves to the cytoplasm, where the hydrolysis of Ran GTP to Ran GDP results in the release of the cargo [29,30]. The presence of F-actin in the nucleus is based on indirect and direct evidence. The indirect evidence includes studies showing that Fragmin, which restricts the polymerization of actin filaments, inhibits transcription of salamander lampbrush chromosomes, and the antibodies against nuclear motor protein myosin-I, which binds the barbed end of actin filaments, inhibit the transcription [16,31]. Among the direct evidence are studies showing a transient presence of F-actin in the nuclei of fibroblasts [32], fibrosarcoma cells [33], breast epithelial [34], and early embryonic mouse cells [35]. Studies also show that during the exit from the mitotic division, the F-actin forms in the nuclei of the daughter cells, where it promotes enlargement of the nucleus, chromatin decondensation, and the formation of nuclear protrusions [32]. The polymerization of nuclear F-actin filaments was also observed during cell spreading and stimulation by fibronectin via interaction with the extracellular matrix [36]. Among the proteins which regulate actin is cofilin, which accelerates actin filament polymerization/depolymerization dynamics [37]. Cofilin severs actin filaments and increases the number of free ends to which actin monomers can be added or removed [37]. Another protein is the actin-monomer-binding protein profilin, which is necessary for the formation of actin cables [38], and fascin, which is required for actin filament bundling [39]. During the stress, cofilin is actively imported into the nucleus using its nuclear localization signal (NLS). Cofilin NLS is importin-α/β-dependent. The interaction of cofilin NLS with the adapter protein importin α and nuclear transport protein importin β transports cofilin through the nuclear pore complex (NPC) into the nucleus. The directionality of the transport and cofilin release in the nucleus is regulated by the GTP-binding nuclear protein Ran (Figure 1). Cofilin also plays a role in the import of actin into the nucleus [40,41,42]. Actin does not have NLS in its sequence, but interestingly, Wada et al. [43] identified in actin’s sequence two nuclear export signals (NES1 and NES2) [43]. Thus, actin entry into the nucleus has to be facilitated by the proteins containing NLS, such as cofilin [42]. The export of nuclear actin is facilitated by the Exportin 6 and the actin-binding protein Profilin (Figure 1) [20,25,44]. Many studies have shown that, in contrast to the cytoplasm, the intranuclear actin does not form long F-actin filaments but short oligomeric filaments. This is possibly to prevent interference with the chromatin [45,46,47] or prevent inhibition of transcription [48]. The field emission scanning electron microscopy (feSEM) studies of *Xenopus* oocytes [47] showed that the nucleus contains a network of short but highly branched, forked, and bundled actin filaments, which are connected to the nuclear pore complexes at one end, and the subnuclear organelles such as Cajal bodies and snurposomes (both involved in the assembly of ribonucleoproteins and mRNA processing), and nucleoli, at the other end. These nuclear pore-linked filaments (PLFs) are 12–100 nm diameter and contain actin, and actin-scaffolding protein 4.1. Because a single actin filament is 8–9 nm in diameter, the intranuclear filaments must also contain other bounds or transiently interacting proteins of unknown identity. The probable candidates could be the actin-interacting proteins identified in the nuclei of *Drosophila* and human cells, such as the ubiquitous protein EAST, the α-actinin-like actin-binding domain-containing protein NUANCE, and the nucleoporins [47,49,50]. Interestingly, in *Xenopus* oocyte nuclei, protein 4.1 is regularly spaced on the filaments at ~120 nm intervals, and the pairs of protein 4.1-containing speckles are present at the filament forks [47,51]. Besides *Xenopus*, the presence of protein 4.1 on the intranuclear filaments was also observed in the nuclei of human fibroblasts [47]. In human erythrocytes, the protein 4.1, also known as the Beatty’s Protein, regulates the stability of the erythrocyte membrane through its interaction with the short actin filaments and spectrin [52]. It seems that some of the functions of intranuclear actin filaments could be the facilitation of intranuclear transport between the nuclear pores and subnuclear organelles, and a displacement of the membrane-underlying chromatin to form the chromatin-free spaces under the nuclear pores, which will facilitate the transport of the molecules [47,53]. Some of the existing techniques of actin visualization in live cells involve the tagging of actin with fluorescent proteins or the expression of GFP-actin [54]. However, the comprehensive analysis of nuclear actin functions will require the development of novel, super-resolution imaging techniques. Recently developed multiplexed super-resolution volumetric imaging and expansion microscopy (ExM) enable visualization of nanoscale details of actin filament organization [55].

## 3. Enigmatic Role of Intranuclear Actin Rods

Actin rods are several μm-long aggregates of actin filaments and actin-interacting proteins [56,57]. Because they form in the nucleus and/or cytoplasm in response to mechanical or chemical/environmental stress (heat shock, hypoxia, pathogens, toxin, drugs), the actin rods are the marker of the so-called Actin Stress Response (ASR) that correlates with many human diseases. The intranuclear rods were first discovered around 1980 in the dimethyl sulfoxide (DMSO)-stressed *Dictyostelium* amoeba, human HeLa, and fetal lung WI-38 cells, and rat kangaroo kidney epithelial PtK2 cells [58,59]. Although the function of the intranuclear rods is not clear, it seems that they may act as a protective mechanism by eliminating actin-treadmilling and, thus, frees up ATP that can be utilized for the immediate needs of stressed cells [56,60,61]. Several actin-interacting proteins have been identified in the intranuclear rods [56]. One of these proteins is Cofilin, which in the nuclear rods of some, but not all cells, binds the whole length of actin filaments, which makes the rods undetectable by a routine actin staining by phalloidin [62]. Studies on *Drosophila* heat shock model showed that the formation of intranuclear actin rods requires an increase in the nuclear pool of free actin and a heat-induced increase in the activity of cofilin [63,64]. Studies on the nuclear rod formation in *Drosophila* ovaries showed that cofilin is regulated by nuclear actin-bundling protein fascin [65].

Another protein present in the actin rods is the Actin-interacting protein 1 (Aip1), a WD repeat-containing protein that facilitates the disassembly of actin filaments [66]. Other proteins are the actin-binding protein Coronin (CorA), which also belongs to the WD-repeat family of proteins and is involved in actin cytoskeleton organization [67], and the calcium-regulated Actin-bundling protein B (AbpB) [56]. The nuclear rods of *Dictyostelium* contain the actin variant Filactin (Fia) [56]. Ishikawa-Ankerhold, et al. [56] described the spatiotemporal sequence of the formation of intranuclear rods and recruitment of its components in DMSO-stressed *Dictyostelium.* The formation of the rod starts 5 min after stressor application and lasts 30–60 min. During the first 5–10 min, the actin aggregates with the cofilin into short spikes in the vicinity of the nuclear envelope. Between 15 and 20 min, the spikes recruit Fia and Aip1 and aggregate into bundles, and between 30 and 60 min, the bundles associate with AbpB and CorA, and aggregate into mature, thick rods. The knockout studies showed that while the Aip1 is necessary, the Fia and CorA are not essential for the formation of the rods. After removal of the stressor, the rods were disassembled within 30 min. The disassembly was delayed in the absence of CorA or Fia, and faster in the absence of Aip1, which was expected because the Aip1-deficient cells did not form fully developed rods [56].

There are indications that the intranuclear rod formation during the stress is induced by the proteolytic fragments of small GTPase RhoA [68]. RhoA is a major regulator of actin polymerization and, thus, actin-related cell functions [69]. Under stress conditions, depending on the type of stressor, RhoA can be either activated or downregulated by the proteolytic cleavage by calpain, caspases, and serine proteases. The amino-terminal cleavage fragment (RhoA-NTF) or carboxy-terminal fragment (RhoA-CTF), which form after the exposure of fibroblasts to H_2_O_2,_ induce the actin stress fibers in the cytoplasm, while RhoA-CTF also induces the cytoplasmic actin rods in the vicinity of the nuclear membrane [68,70]. This suggests that RhoA-CTF may also translocate to the nucleus and promote intranuclear rod formation [68].

It should be noted that the inability of the cell to disassemble actin rods can be highly toxic and correlates with many human diseases [71,72]. Although it is often unknown if the nuclear/cytoplasmic actin rods are the causative agent or the byproduct of the disease, there are a hallmark of Alzheimer’s, various neurodegenerative diseases, Huntington’s disease and nemaline myopathy [62,73,74,75,76]. The pathology of Alzheimer’s brains shows extracellular plaques of β-amyloid (Aβ) peptides. The soluble form of Aβ causes activation of cofilin, which promotes the formation of actin rods. Rods accumulate in neurites and cause synaptic dysfunction [62]. Similarly, in Huntington’s disease, the inability to clear the nuclear rods from the nerve cells may lead to disease development and progression [74,75]. The intranuclear actin rods are also present in many forms of congenital myopathy, which are caused by the mutation in the α skeletal actin (ACTA1) gene. The mutated ACTA1 induces intranuclear actin rods, which are aggregated through the interaction with α actinin. The accumulation of intranuclear rods in the muscle cells causes their dysfunction and muscle weakness [77]. Because of these findings, actin rods are a potential novel therapeutic target in many human diseases.

## 4. Actin Role in Nuclear Membrane Rupture

In the majority of dividing cells, the nuclear envelope breaks before the division, allowing the attachment of the chromosomes to the division spindle [78]. The nuclear envelope consists of the outer and inner membrane, nuclear pore complexes, and nuclear lamina, which underlines the inner membrane and is composed of a variety of the type V intermediate filament proteins, called nuclear lamins [79]. In humans, the mutations in nuclear lamins cause many hereditary diseases, including muscular dystrophy, peripheral neuropathy, and progeria [80]. The first step in the nuclear envelope breakdown (NEBD) is the disassembly of the nuclear pore complexes. This is followed by the depolymerization of the nuclear lamina, and, thus, the weakening of the nuclear envelope [81,82]. Common belief has been that the mechanical force needed for the final breakage of the nuclear envelope comes from the microtubules, which after attaching, through the motor protein dynein, to the weakened nuclear envelope, pull and break the membrane [83,84]. Although this is certainly true for small-size somatic cell nuclei, recent studies showed that in the large-size oocyte nuclei, instead of the microtubules, actin is very much involved in the breakage of the nuclear envelope [85]. Studies on starfish oocytes showed that, before the NEBD, an Arp2/3 complex-dependent polymerization of actin leads to the formation of a shell consisting of densely packed branched actin within the nuclear lamina (Figure 2) [86,87]. Subsequently, the actin spikes, protruding from the actin shell, pierce and fragment the nuclear membrane while leaving the nuclear lamina intact (Figure 2) [87,88]. Because the actin shell is also present in the nuclei of oocytes and dividing embryos in other echinoderm species, it seems that the actin-aided rupture of the nuclear envelope is, if not universal, much more common than previously thought [85].

## 5. Actin Role in Chromatin Organization and Remodeling

A single human cell contains around 2m of DNA that is packed into a ~10μm nucleus. To achieve this, the DNA is wounded around histones and compacted into the heterochromatin [89]. Because such condensed DNA is unavailable to the transcription factors, it has to undergo remodeling into the more open, transcription factors-accessible euchromatin [90]. The nucleus contains chromatin and the interchromatin compartments [91,92]. In the interphase nucleus, the individual chromosomes occupy their designated place—a chromosome territory [93]. Thus, chromatin is not randomly distributed within the chromatin compartment but is sub-compartmentalized into defined and functionally different domains, whose chromatin is anchored to the nuclear lamina, nucleoli, or other subnuclear bodies. One of the functions of chromatin/chromosome compartmentalization is the inclusion or exclusion of the specific factors involved in the regulation of gene splicing, silencing, transcription, and replication [94]. Chromatin compartmentalization is accompanied by the compartmentalization of nuclear actin [91]. In the interphase nucleus, chromatin displays constant movements and reordering, which allow for repositioning of the loci and the placement of the transcriptionally active genes in a common nuclear space, where they can be co-transcribed [93]. Time-lapse observations of the fluorescently tagged loci or chromatin territories showed that in mammalian cells, the chromatin is more mobile in the early G1 phase than in the S or G2 phase of the cell cycle. These studies showed that in early G1 the chromatin territories move between 0.47 and 4.44 μm, and only 0.25–2.11 μm in the later stages [95,96]. There are two major mechanisms of chromatin mobility, one is the Brownian’s motions, and another is an active, actin-dependent, and ATP-dependent movement, which allows reordering chromatin compaction to access genetic loci for transcription or repair. Actin functions in this process as a subunit of the chromatin-modifying (remodeling) complexes (Figure 3) [16,97].

Among many actin-containing chromatin remodeling complexes in yeast, invertebrate, and vertebrate cells, the most studied are BAF, INO80, NuA4/TIP60, SWR1, and Mi-2 [16,98,99]. BAF, the ATP-dependent Brahma-related gene (Brg)/Brahma-associated factor, also called the SWItch/Sucrose Non-Fermentable (SWI/SNF) is critical for lineage specification in the early mouse embryo and neuronal development in humans [98,100]. Mutations of the BAF subunits are linked to autism, schizophrenia, and many neurodevelopmental disorders [100]. INO80 complex contains the Ino80 ATPase that is a member of the SNF2 family of ATPases [101]. It is possible that in ATP-dependent chromatin remodeling complexes such as INO80, ATP binding to actin results in the conformational change of the whole complex allowing it to remodel. These complexes, using the energy produced by ATP hydrolysis, also participate in the movement of nucleosomes and the chromatin [102]. These complexes use the ATP-derived energy to displace or exchange, package, and position the histones/or histone variants within the nucleosomes [16,103,104,105,106,107,108]. On the other hand, the monomeric actin found in the INO80 complex regulates its ability to bind and mobilize nucleosomes and repair DNA damage. It has been suggested that the actin-Arps that modulates INO80 is also present in other remodeling complexes serving as a platform to contact the nucleosomal DNA [16]. Additionally, the analysis of the crystal structure of the Arp8 module of the *Saccharomyces cerevisiae* INO80, showed that during chromatin remodeling, the actin-Arps complex facilitates recognition of the extranucleosomal 40-bp linker DNA [109,110]. Two different functions of actin in the chromatin remodeling complexes have been proposed. Either the complex is attached to the actin filament and moves along the filament propelled by the filament polymerization-depolymerization, or the monomeric actin directly interacts with the chromatin, or other unconventional forms of actin oligomers, performing some (currently unknown) functions in the chromatin remodeling complexes [16]. Another chromatin-modifying complex is the histone acetyltransferase complex NuA4/TIP60, which is involved in chromatin-binding, histone modifications, transcription, DNA damage response, control of the cell cycle, nonhomologous end joying, and homologous recombination [16,111,112,113]. Recent studies showed that the SWR1 remodeling complex deposits Htz1 histone into the chromatin and prevents genome instability, which is often associated with tumorigenesis [114,115]. The incorporation of Htz1 histone into the chromatin may stabilize damaged replication forks (by preventing fork regression), or if the replication fork had collapsed, the Htz, incorporated at the sites of replication damage, prevents DNA double-strand breaks [115]. The Mi-2/NuRD (Nucleosome Remodeling Deacetylase) complex has ATP-dependent chromatin remodeling activity and also histone deacetylase activity [99]. Studies on *Drosophila* showed that the chromatin containing DNA with the double-strand breaks relocates to the nuclear periphery for repair. The relocation of this damaged DNA occurs through a myosin-dependent sliding on the actin filaments, which polymerase in response to the DNA damage [110,116]. Recent studies using Xenopus cell-free extracts, human bone osteosarcoma epithelial cells U2OS, and mouse-tail fibroblast cell lines showed that nuclear actin together with Arp2/3 and WASP are recruited into the chromatin undergoing the homology-directed repair (HDR) [117]. These studies also showed that actin repositions DNA containing the double-strand brakes (DSBs) undergoing HDR, into the specific chromatin repair domains and that the inhibition of actin nucleation decreases HDR efficiency [117]. Another study on Xenopus cell-free extracts and human somatic cells showed that DNA replication (both initiation and elongation) depends on nuclear shuttling dynamics of actin and formin [22,118]. Live and supper-resolution imaging studies of human lung fibroblast cell line IMR90 showed that during replication stress (induced by DNA polymerase inhibitor aphidicolin or the ribonucleotide reductase inhibitor hydroxyurea), the replication foci are translocated along actin filaments to the nuclear periphery where they undergo repair [119].

## 6. Actin Role in Transcription

Besides being a component of chromatin remodeling complexes, nuclear actin regulates various transcription factors and associates with all RNA polymerases [120,121], i.e., RNA polymerase I that transcribes rRNA, RNA polymerase II that transcribes mRNA, miRNA, snRNA, and snoRNA, and RNA polymerase III that transcribes tRNA and 5S rRNA [26,120,121,122]. The interaction of actin with these three RNA polymerases is mediated by the RNA-binding proteins, Rbp6 and Rbp8 [121]. Both G-actin and actin oligomers are known to associate with RNA polymerases [122,123]. The in vivo and in vitro studies of the Pol I-, Pol II-, and Pol III-dependent transcription [121] showed that actin is recruited into the gene promoters, where it recruits polymerases, becomes a part of the polymerase preinitiation complexes (Figure 3), and together with the nuclear myosin-1 is involved in transcript elongation [99]. Recent studies using RNA-seq and super-resolution imaging [120] showed that the serum stimulation and interferon-γ treatment induce the assembly of nuclear actin filaments, which in turn, enhance the clustering of polymerase II (Pol II) and promote transcription of serum- and interferon-γ-inducible genes in the CRISPR-edited osteosarcoma U2OS cell line [120]. However, it is still unknown how nuclear actin assembles the clusters of Pol II on the specific genes [120].

Actin also directly interacts with the heterogeneous nuclear ribonucleoprotein U (hnRNP U), which contains the actin-binding site in its C-terminus and is a component of pre-mRNA particles [99]. Studies of the conditional knockout of the Hnrnpu gene in the mouse heart showed that the hnRNP U is required for mRNA splicing and heart development [124]. Studies of the transcription process in the dipteran *Chironomus* showed that actin binds directly to the hnRNP, HRP65-2 that facilitates the recruitment of H3-specific acetyltransferase p2D10 [99,121,122,125]. After transcription, actin, incorporated into the hnRNPs, moves with the newly synthesized transcript to the polyribosomes [110].

Because the changes in the chromatin and genome architecture are also known attributes of the embryonic and post-embryonic cellular differentiation and correlate with many different diseases, one of the fascinating roles of nuclear actin is its involvement in the establishment of cell fate, differentiation programs during neurogenesis, myogenesis, organs’ development, and development of various diseases [110,124].

## 7. Nuclear Actin in Macrophages and Other Immune Cells

The multifunctionality of nuclear actin suggests that nuclear actin has to be also important for the differentiation and functions of immune cells. For over a decade, our laboratory has studied the structure and functions of macrophages in the rejection of transplanted organs. Our studies in rodent transplantation models showed that the integrity of macrophage actin cytoskeleton is indispensable for the macrophage-dependent long-term phase of graft rejection [126,127,128]. These findings also prompted our interest in the role of nuclear actin in the macrophages and the immune cells, in general. In the following paragraph, we describe what is currently known about nuclear actin in macrophages and other immune cells.

During the activation of the immune response, the monocytes circulating in the blood are recruited by the inflammatory signals to the site of inflammation or infection, where they differentiate into the macrophages. In the *in vitro* experiments, the blood-derived monocytes or bone marrow-derived cells can be forced to differentiate into macrophages, and/or polarized into different macrophage subtypes by the addition of various cytokines and factors, such as, mentioning a few, interferon-gamma, interleukins, the bacterial cell wall component lipopolysaccharide (LPS) or phorbol 12-myristate 13-acetate (PMA) [126,127,128]. Interestingly, the LPS has been shown to reorganize actin cytoskeleton in many different cell types, including macrophages [129]. Studies of human blood-derived monocytes and promyelocytic leukemia cells (HL-60) showed that during the PMA-induced differentiation of these cells into the macrophages, there is a massive influx, dependent on the p38 mitogen-activated kinase, of actin into the cell nucleus. Nuclear actin content increased 12–32-fold during 24–72 h of PMA treatment, and the recruited nuclear actin was incorporated into Pol-II complexes [130,131]. The ChiP-on-chip experiments identified the actin-bound promoters of the genes related to the chromatin remodeling, transcription, RNA splicing, apoptosis, and immune response, and the knockout experiments showed that actin binding to the promoters modulated the expression of these genes [130]. Studies on the migration of keratinocytes showed that nuclear actin regulates transcription of genes related to the focal adhesions and cell migration and that the knockout of the importin 9, which lowered nuclear actin level, increased keratinocyte migration [132]. These studies are also relevant to the macrophages and other immune cells, which have to actively migrate to the target tissues and organs.

Studies of the nuclear signaling events during CD4^+^ T cell activation showed that the T cell antigen receptor (TCR) signaling, which drives T cell differentiation and proliferation, induces the formation of the nuclear actin network. Nuclear actin regulates the expression of IL-2, IL-6, IL-9, IL-10, IL-21, IFN-γ, and TNF-α, thus, the immune functions of the T cells [133]. Additionally, the actin-dependent recruitment of RNAPII at promoter sites regulates the transcription of genes involved in T cell differentiation [134].

All these studies indicate that the drugs modulating the nuclear/cytoplasmic actin dynamics in the immune cells could be potentially used as an anti-inflammatory and/or anti-cancer therapies [126,127,135].

Taking into consideration the multifaceted role of nuclear actin and a dynamic exchange between the nuclear and cytoplasmic actin pool, the future challenge will be to further define the functions of nuclear actin in various cellular processes and different cell types and organisms.

## Figures and Tables

**Figure 1 biology-10-00304-f001:**
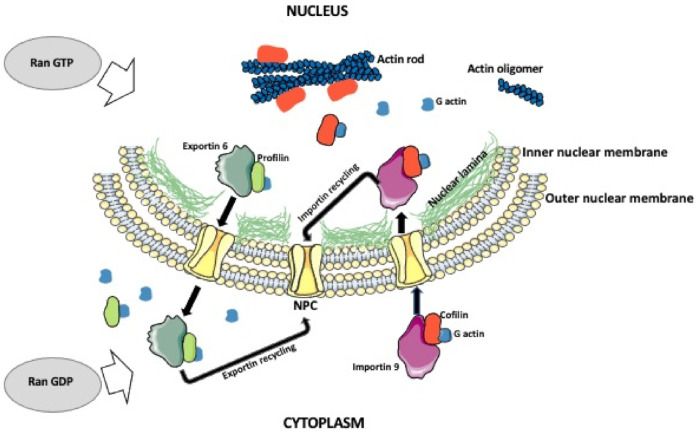
Nuclear actin shuttling and actin rode formation. Actin monomers in the cytoplasm form a complex with cofilin and importin 9. The complex is imported into the nucleus, where actin is released, and the importin is cycled back to the cytoplasm. The monomeric nuclear actin can form oligomers, polymers, or actin rods. The export of the actin from the nucleus is facilitated by the profilin and exportin 6. While in the cytoplasm, exportin is recycled back into the nucleus. This process depends on the balance between nuclear and cytoplasmic Ran GTP/GDP. Ran GTP promotes nuclear export and Ran GDP promotes nuclear import.

**Figure 2 biology-10-00304-f002:**
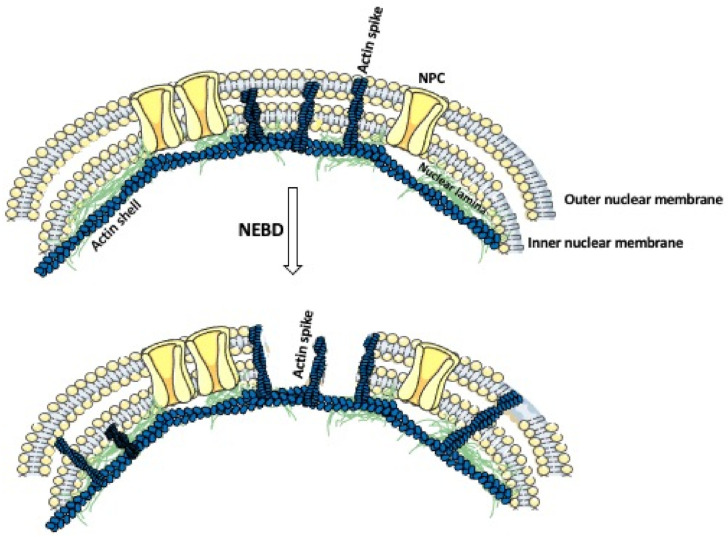
Actin’s role in the nuclear envelope breakdown. The nuclear envelope is composed of a double nuclear membrane and an underlying nuclear lamina. Prior to the nuclear envelope breakdown (NEBD), the actin shell polymerizes within the nuclear lamina. This is followed by the formation of the actin spikes, which displace the nuclear pore complexes (NPCs), and penetrate and fragment the nuclear membrane.

**Figure 3 biology-10-00304-f003:**
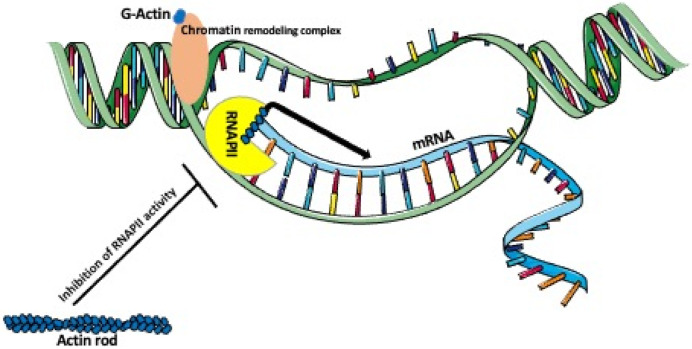
Actin’s role in chromatin remodeling and transcription. Nuclear G-actin, as a part of a chromatin remodeling complex, participates in remodeling a condensed transcriptionally inactive heterochromatin into the RNA polymerase accessible decondensed euchromatin. Monomeric or oligomeric actin forms a complex with RNAPII promoting the enrichment of RNPII at the promoter and the transcription of mRNA. In contrast, nuclear actin rods, which are formed during the stress, inhibit the actin-dependent activity of RNAPII.

## Data Availability

Not applicable.

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
