# Peer review of "New Insights into Cellular Functions of Nuclear Actin"

_biology, 2021, doi:10.3390/biology10040304_

Round 1
Reviewer 1 Report
This review by Kloc et al. aims to discuss new advances made in the nuclear actin field and their relevancy to human health and disease. The “nuclear actin field” has significantly developed over the last decade due to the designment of probes that label specifically nuclear actin and to advances made in microscopy techniques. Thus, this is an interesting and timely review. It is generally clear and well written. I have two major concerns: 1. The structure of the review is confusing. The authors combined sections describing different actin polymerization states (“intranuclear actin network” and “intranuclear actin rods”) with sections describing specific roles of actin in the nucleus (“Actin ruptures the nuclear membrane” and “actin role in chromatin organization and remodeling”). Both structures are acceptable, but it would make the review easier to read if the authors were consistent with one. Either subheading according to the polymerization state or according to the role in the nucleus. 2. The authors missed several major advances that were made recently and described the role of nuclear actin in the repair of DNA damage and in replication dynamics. For example: • Parisis N, Krasinska L, Harker B, Urbach S, Rossignol M, Camasses A, et al. Initiation of DNA replication requires actin dynamics and formin activity. EMBO J. • Lamm N, Read MN, Nobis M, Van Ly D, Page SG, Masamsetti VP, et al. Nuclear F-actin counteracts nuclear deformation and promotes fork repair during replication stress. Nat Cell Biol. • Schrank BR, Aparicio T, Li Y, Chang W, Chait BT, Gundersen GG, et al. Nuclear ARP2/3 drives DNA break clustering for homology-directed repair. Nature. These works were published in high impact factors journals and should be discussed.Author Response
Review 1
This review by Kloc et al. aims to discuss new advances made in the nuclear actin field and their relevancy to human health and disease. The “nuclear actin field” has significantly developed over the last decade due to the designment of probes that label specifically nuclear actin and to advances made in microscopy techniques. Thus, this is an interesting and timely review. It is generally clear and well written. I have two major concerns:
- The structure of the review is confusing. The authors combined sections describing different actin polymerization states (“intranuclear actin network” and “intranuclear actin rods”) with sections describing specific roles of actin in the nucleus (“Actin ruptures the nuclear membrane” and “actin role in chromatin organization and remodeling”). Both structures are acceptable, but it would make the review easier to read if the authors were consistent with one. Either subheading according to the polymerization state or according to the role in the nucleus.
Response: As requested we renamed the subheads
- The authors missed several major advances that were made recently and described the role of nuclear actin in the repair of DNA damage and in replication dynamics. For example: • Parisis N, Krasinska L, Harker B, Urbach S, Rossignol M, Camasses A, et al. Initiation of DNA replication requires actin dynamics and formin activity. EMBO J. • Lamm N, Read MN, Nobis M, Van Ly D, Page SG, Masamsetti VP, et al. Nuclear F-actin counteracts nuclear deformation and promotes fork repair during replication stress. Nat Cell Biol. • Schrank BR, Aparicio T, Li Y, Chang W, Chait BT, Gundersen GG, et al. Nuclear ARP2/3 drives DNA break clustering for homology-directed repair. Nature. These works were published in high impact factors journals and should be discussed.
Response: As requested we added these references and discussed the findings

Reviewer 2 Report
Re: Ms 1152264
This manuscript is a timely and informative compendium on almost everything known today
about nuclear actin – its existence, its structural identification as G- or F-actin and as compact
rods, its mobility and its p roven or assumed functions. As stated in the introduction, nuclear
actin has quite a history from being denied its existence up to an intensily investigated,
multifunctional and vital nuclear component. The presented manuscript may therefore serve
as an important source of information on many nuclear activities depending on actin, but its
presentation should be improved prior to print, as follows:
- In the para on antibodies detecting nuclear actin, the mouse monoclonal antibody 2G2
is introduced as raised against Xenopus actin. This is not correct: it was generated
against vertebrate skeletal muscle actin, recognizing nuclear actin in many different
species, including Xenopus.
- In some parts, citation of references is incomplete. Examples: (i) Ref. 6 should be cited
as the correct reference revealing the identification of the 2G2 binding epitope. (ii) The
amount of nuclear actin is given to 20% of total cellular actin. Where does this amazing
figure come from?
- The ms suffers from some problems of the (mostly Polish?) authors with the English
language. Reading it would definitely be easier after some corrections, mainly in the
use of articles. I enclose a version where I suggested some improvement.
Author Response
Review 2
Re: Ms 1152264
This manuscript is a timely and informative compendium on almost everything known today
about nuclear actin – its existence, its structural identification as G- or F-actin and as compact
rods, its mobility and its proven or assumed functions. As stated in the introduction, nuclear
actin has quite a history from being denied its existence up to an intensily investigated,
multifunctional and vital nuclear component. The presented manuscript may therefore serve
as an important source of information on many nuclear activities depending on actin, but its
presentation should be improved prior to print, as follows:
- In the para on antibodies detecting nuclear actin, the mouse monoclonal antibody 2G2
is introduced as raised against Xenopus actin. This is not correct: it was generated
against vertebrate skeletal muscle actin, recognizing nuclear actin in many different
species, including Xenopus.
Response: thank you for pointing this out, we corrected this statement
- In some parts, citation of references is incomplete. Examples: (i) Ref. 6 should be cited
as the correct reference revealing the identification of the 2G2 binding epitope.
Response: we corrected the reference citation
(ii) The amount of nuclear actin is given to 20% of total cellular actin. Where does this amazing
figure come from?
Response: thank you for catching this error, we deleted this sentence
- The ms suffers from some problems of the (mostly Polish?) authors with the English
language. Reading it would definitely be easier after some corrections, mainly in the
use of articles. I enclose a version where I suggested some improvement.
Response: We could not find any attachment with your suggested corrections on the Website, so the manuscript was corrected by the native English colleague. Of course, if you provide it we will be happy to introduce the modifications.

Reviewer 3 Report
This is a review on “nuclear actin”. One of many many reviews and few original research papers (some of them not even cited in this manuscript).
Points of criticisms:
-What means : “nuclear actin assumes 13 many different conformations “ I am not aware of many or any different conformations. There is a lack of evidence for that. Older papers from the 80ties have not been reproduced and are like due to antibody issues. The beta actin inside the nucleus is the same as in the cytoplasm. Unless the authors talk about Aprs.
-the term “nuclear actin rods” should be avoided. Or clearly defined and separated from other F-actin structures. There is now numerous evidence that physiological actin filaments form (Baarlink et al., Science 2013; Caridi et al Nature 2018; Wang, Sherrard et al., Nature Communications 2019).
-the authors sate: “After decades of skepticism and denial, at last, around ten years ago, ….” And then cite a review as evidence? They should cite original papers and real fits evidence for actin polymerisation in living cells. They should also define the cell system then.
-The go on : “One of the reasons for the nuclear actin presence denial was a frequent inability to 29 visualize ….” To my knowledge the first visualisation due to the first usage of nuclear targets probes is from Baarlink et al., 2013, Science. That landmark paper is not even cited.
-the review generally cites too many reviews. Already from the first 20 references 8 are reviews!The authors should revise this. It reads like a meta-review.
-in the chapter intranuclear actin rods they cite ancient and dubious literature that intoxicated cells (with DMSO for example) to draw conclusions on nuclear events? How can this be helpful for our understanding today?
-they draw an actin rod as a F-actin bundle….is there any real evidence for nuclear actin bundling? And if so, through which factors?
- the chapter “Actin ruptures the nuclear membrane” about NEBD is not really about actin inside the nuclear compartment. Figure 2 seems wrong. To my knowledge this event is caused by a perinuclear actin shell and not from a actin ring at the inner nuclear membrane. This figure needs to be revised.
-Figure 3 shows a actin rod again- what exactly is meant here? There is nice novel work using super resolution imaging showing F-actin patches near RNA-PolII (Wei et al., Science Advances , 2020), which the authors even cite. Or do they refer to older work from the Grummt group?
-Did the authors cite the papers on CENP-A loading and nuclear actin and mitosis? I could not finde these important papers. Instead the review is full of literature in before-modern-imaging-times from 30-40 years ago. There are mitosis papers on nuclear actin missing (Parisis et al., Embo J, 2017; Krippner et al., Embo reports 2020).
Author Response
Review 3
What means : “nuclear actin assumes 13 many different conformations “ I am not aware of many or any different conformations. There is a lack of evidence for that. Older papers from the 80ties have not been reproduced and are like due to antibody issues. The beta actin inside the nucleus is the same as in the cytoplasm. Unless the authors talk about Aprs.
Response: Such a sentence does not exist in the manuscript, the number “13” is the line # of the sentence, so the sentence reads: “nuclear actin assumes many different conformations”. We changed the word “conformations” to “pools”. However, we do not agree that there are no proofs of different conformations (besides typical G and F actin) of nuclear actin. Recent research data confirm that there are multiple pools of nuclear actin. These different pools of nuclear actin are recognized by different antibodies (which very strongly suggests a different conformation of actin), have different localization within the nucleus, and different abundance at different stages of cell cycle or development. There are also many studies, which show that the intranuclear actin does not form long filaments (F-actin) but, possibly to prevent interference with the chromatin, it remains as short oligomeric filaments. For example, the nuclear actin oligomers, which can’t be stained by phalloidin, are recognized by specific antibodies such as the 2G2 antibody, which recognizes the epitope composed of three nonsequential regions (aa131-139, aa155-169, and aa176-187) located in the vicinity of the nucleotide-binding region. This indicates that the nuclear actin adopts a specific conformation, absent in the cytoplasmic F-actin, which compacts these three separate regions into a single antibody-recognizable epitope. One of such recent reports is a very thorough study from 2018 on the multiple pools of nuclear actin during Drosophila oogenesis: Ref# 8. Wineland DM, Kelpsch DJ, Tootle TL. Multiple Pools of Nuclear Actin. Anat Rec (Hoboken). 2018 Dec;301(12):2014-2036. doi: 10.1002/ar.23964. Addionally, as described by Kelpsch and Tootle, 2018, “the nuclear actin polymers are oligomers of actin that do not have an obvious filament structure, while nuclear actin rods are larger polymers of actin that resemble either cytoplasmic actin filaments or bundles”
-the term “nuclear actin rods” should be avoided. Or clearly defined and separated from other F-actin structures. There is now numerous evidence that physiological actin filaments form (Baarlink et al., Science 2013; Caridi et al Nature 2018; Wang, Sherrard et al., Nature Communications 2019).
Response: We do not agree that the term “nuclear actin rods” should be avoided or eliminated. The presence of these structures is well documented by many studies. The rods are present not only in cells exposed to mechanical or chemical/environmental stress but also present in many physiological and pathological conditions, and are a hallmark of Alzheimer’s, various neurodegenerative diseases, nemaline myopathy, and Huntington’s disease. We believe that the fact that we created a separate subchapter to describe the nuclear rods indicates that we clearly separated them from other F-actin structures. We also cited already the Baarlink and Caridi papers indicated by the reviewer.
-the authors sate: “After decades of skepticism and denial, at last, around ten years ago, ….” And then cite a review as evidence? They should cite original papers and real fits evidence for actin polymerisation in living cells. They should also define the cell system then.
Response: as requested we added the original references (total 9 references) and cell systems
-The go on : “One of the reasons for the nuclear actin presence denial was a frequent inability to 29 visualize ….” To my knowledge the first visualisation due to the first usage of nuclear targets probes is from Baarlink et al., 2013, Science. That landmark paper is not even cited.
Response: We have already cited 3 different and more recent Baarlink papers, which all cite Baarlink, 2013 paper
-the review generally cites too many reviews. Already from the first 20 references 8 are reviews! The authors should revise this. It reads like a meta-review.
Response: as requested we added more original papers
-in the chapter intranuclear actin rods they cite ancient and dubious literature that intoxicated cells (with DMSO for example) to draw conclusions on nuclear events? How can this be helpful for our understanding today?
Response: This is incorrect, we cite many new papers published in very well-known and established journals:
Ishikawa-Ankerhold, H., Daszkiewicz, W., Schleicher, M. et al. Actin-Interacting Protein 1 Contributes to Intranuclear Rod Assembly in Dictyostelium discoideum. Sci Rep. 2017; 7, 40310. https://doi.org/10.1038/srep40310
Kelpsch DJ, Tootle TL. Nuclear Actin: From Discovery to Function. Anat Rec (Hoboken). 2018 Dec;301(12):1999-2013. doi: 10.1002/ar.23959.
Munsie LN, Desmond CR, Truant, R. Cofilin nuclear-cytoplasmic shuttling affects cofilin-actin rod formation during stress. J Cell Sci. 2012 Sep 1;125(Pt 17):3977-88. doi: 10.1242/jcs.097667.
Figard L, Zheng L, Biel N, Xue Z, Seede H, Coleman S, Golding I, Sokac AM,Cofilin-Mediated Actin Stress Response Is Mal-adaptive in Heat-Stressed Embryos, Cell Reports, 2019, 26, 3493-3501.e4, https://doi.org/10.1016/j.celrep.2019.02.092.
Biel, N., Figard, L., Sokac AM. Imaging Intranuclear Actin Rods in Live Heat Stressed Drosophila Embryos. J Vis Exp. 2020, 159:10.3791/61297. doi: 10.3791/61297. PMID: 32478727; PMCID: PMC7521865.
Serebryannyy LA, Yuen M, Parilla M, Cooper ST, de Lanerolle, P. The Effects of Disease Models of Nuclear Actin Polymeri-zation on the Nucleus. Front Physiol. 2016 Oct 7;7:454. doi: 10.3389/fphys.2016.00454.
Bamburg JR, Bernstein BW. Actin dynamics and cofilin-actin rods in Alzheimer disease. Cytoskeleton (Hoboken). 2016 Sep;73(9):477-97. doi: 10.1002/cm.21282.
Domazetovska A, Ilkovski B, Cooper ST, Ghoddusi M, Hardeman EC, Minamide LS, Gunning PW, Bamburg JR, North KN. Mechanisms underlying intranuclear rod formation. Brain. 2007 Dec;130(Pt 12):3275-84. doi: 10.1093/brain/awm247.
And, also as we stated above, the presence of nuclear actin rods is well documented by many studies. The rods are present not only in cells exposed to mechanical or chemical/environmental stress but also in many physiological and pathological conditions, and are a hallmark of Alzheimer’s, various neurodegenerative diseases, nemaline myopathy, and Huntington’s disease.
-they draw an actin rod as a F-actin bundle….is there any real evidence for nuclear actin bundling? And if so, through which factors?
Response: we have already described the bundling and the bundling factors: “The formation of the rod starts 5 min after stressor application and lasts 30-60 min. During the first 5-10 min, the actin aggregates with the cofilin into short spikes in the vicinity of the nuclear envelope. Between 15-20 min, the spikes recruit Fia and Aip1 and aggregate into bundles, and between 30-60 min the bundles associate with AbpB and CorA, and aggregate into mature, thick rods”, and also, ” Studies on the nuclear rod formation in Drosophila ovaries showed that cofilin is regulated by nuclear actin-bundling protein Fascin “ and “Other proteins are the actin-binding protein Coronin (CorA), which also belongs to the WD-repeat family of proteins, and is involved in actin cytoskeleton organization, and the calcium-regulated Actin-bundling protein B (AbpB).”
- the chapter “Actin ruptures the nuclear membrane” about NEBD is not really about actin inside the nuclear compartment. Figure 2 seems wrong. To my knowledge this event is caused by a perinuclear actin shell and not from a actin ring at the inner nuclear membrane. This figure needs to be revised.
Response: the reviewer is incorrect, actin shell forms within the nuclear lamina, which underlines the inner nuclear membrane, so the Figure 2 is correct. See “High-resolution light microscopy confirms previous findings by immunogold electron microscopy (EM) that the F-actin shell forms on the inner, nuclear side of the NE” in Ref # 46: Mori M, Somogyi K, Kondo H, Monnier N, Falk HJ, Machado P, Bathe M, Nédélec F, Lénárt, P. An Arp2/3 nucleated F-actin shell fragments nuclear membranes at nuclear envelope breakdown in starfish oocytes. Curr Biol. 2014 Jun 16;24(12):1421-1428. doi: 10.1016/j.cub.2014.05.019 and Ref #47: Wesolowska N, Avilov I, Machado P, Geiss C, Kondo H, Mori M, Lenart, P. Actin assembly ruptures the nuclear envelope by prying the lamina away from nuclear pores and nuclear membranes in starfish oocytes. Elife. 2020 Jan 28;9:e49774. doi: 10.7554/eLife.49774
-Figure 3 shows a actin rod again- what exactly is meant here? There is nice novel work using super resolution imaging showing F-actin patches near RNA-PolII (Wei et al., Science Advances , 2020), which the authors even cite. Or do they refer to older work from the Grummt group?
-Did the authors cite the papers on CENP-A loading and nuclear actin and mitosis? I could not finde these important papers.
Response: In Fig. 3 the actin rods indicate that they inhibit Pol II activity. We also show actin in the vicinity of RNA-Pol II. As for CENP-A loading we did not cover in our review the actin role in mitosis or CENP-A-loading, and we did not cover the CENP-A role at the at centromeres and kinetochore assembly. These subjects are so vast that they would require a separate review
Instead the review is full of literature in before-modern-imaging-times from 30-40 years ago.
Response: We are really surprised by the reviewer extreme bias against the papers published before the modern-imaging-time. In reality, these old papers very often contain the fundamental information, which is the basis for the more current research, and without which the current research would not exists. The papers of DNA and actin discoveries are the example of such before the modern-imaging-time papers. Watson and Crick paper on double helix structure of DNA was published in 1953, and between 1938-1942 Szent-Györgyi and Ilona Banga discovered actin! However, at the same time, the reviewer, actually contradicting himself, asked several times why we did not reference the old original papers.
The reviewer is also incorrect in the statement that majority of papers we cited are 30-40 old: among 87 references cited in our manuscript 31 references were published between 2003/2014, and 21 references were published between 2015/2020.
There are mitosis papers on nuclear actin missing (Parisis et al., Embo J, 2017; Krippner et al., Embo reports 2020).
We did not cite these papers because we did not cover the role of actin in mitosis, which is extremely vast subject, and would require a separate review.
